# Hsa_circ_0006834 represses intrahepatic cholangiocarcinoma proliferation through activating AMPK-mTOR pathway and autophagy via has-miR-637-NGFR network

**Ji Wang**[1,2,3☯], **Xinyue Qi**[4☯], **Wangqian Zhao**[4☯], **Ziyan Hao**[4], **Kehan Wang**[4], **Yuting Li**[4], **Yue Wang**[4,5*], **Yunshan Zhang**[1,2,3,4*]

**1** Department of Oncology, The Second Affiliated Hospital of Soochow University, Suzhou, P.R. China, **2** Cancer institute, Suzhou Medical College, Soochow university, Suzhou, P.R. China, **3** National Biomedical Technology Innovation Center, Suzhou Biomedical Industry Innovation Center, Suzhou, P.R. China, **4** Department of Human Anatomy and Cytoneurobiology, School of Basic Medicine Sciences, Soochow University, Suzhou, P.R. China, **5** Experimental Teaching Center for Clinical Skills, Experimental Center, Medical College, Soochow University, Suzhou, P.R. China

☯ These authors contributed equally to this work.
* yszhang1@suda.edu.cn (YW); yuew@suda.edu.cn (YZ)

## Abstract

Intrahepatic cholangiocarcinoma (iCCA) is a malignancy with difficult treatment and poor prognosis, whose pathogenesis could be associated with the expression patterns of circular RNAs (circRNAs). Here, we aim to investigate the effects and mechanism of hsa_circ_0006834 on iCCA proliferation. At first, hsa_circ_0006834 was proved to suppress iCCA cells proliferation and induce autophagy following the construction of hsa_circ_0006834 vector. And hsa_circ_0006834 is negatively linked with the proliferation, migration and invasion of iCCA cells through autophagy. Subsequently, RNA pull-down and dual-luciferase reporter assays were performed to validate the hsa_circ_0006834-has-miR-637-NGFR regulatory network. It was demonstrated by qPCR and Western blot that hsa_circ_0006834 stimulates the AMPK-mTOR pathway and triggers autophagy via NGFR after si-NGFR was synthesized and transfected into iCCA cells. Ultimately, after the successful knockdown of AMPK, the role of the AMPK-mTOR pathway in hsa_circ_0006834 regulating autophagy and proliferation was verified. Taken together, our findings revealed that hsa_circ_0006834 activates the AMPK-mTOR pathway and autophagy to suppress iCCA proliferation by has-miR-637-NGFR network, indicating the potential of hsa_circ_0006834 as a biomarker for iCCA diagnosis and therapy.

## Introduction

Cholangiocarcinoma (CCA) is an epithelial cell malignancy arising from intra- or extrahepatic bile ducts, classified as intrahepatic, perihilar, and distal CCA [1]. ICCA

**Data availability statement:** All relevant data are within the manuscript and its Supporting Information files.

**Funding:** This work was supported by the National Natural Science Foundation of China (Grant No. 32200110), the Natural Science Foundation of the Jiangsu Higher Education Institutions of China (22KJB230002), Provincial-level talent program for National center of technologyinnovation for biopharmaceuticals (NCTIB2024JS0101). The funders had no role in study design, data collection, analysis, decision to publish, or manuscript preparation.

**Competing interests:** The authors have declared that no competing interests exist.

**Abbreviations:** AMPK, 5'-Adenosine Monophosphate-Activated Protein Kinase; circRNAs, circular RNAs; CCA, Cholangiocarcinoma; ceRNA, Competing endogenous RNA; CCK-8, Cell counting kit-8; FBS, Fetal bovine serum; iCCA, Intrahepatic cholangiocarcinoma; mTOR, Mammalian target of rapamycin; Mut, Mutant; 3MA, 3-methyladenine; NGFR, Nerve growth factor receptor; NC, Negative control; OD, Optical density; qRT-PCR, Quantitative reverse transcription polymerase chain reaction; Rapa, Rapamycin; 3' UTR, 3' untranslated region; WT, Wild-type

occurs in the bile ducts within the liver parenchyma, characterized by late diagnosis and difficult treatment [2]. Although historically iCCA has been treated with surgical resection of early-stage disease and cytotoxic therapy for locally advanced or metastatic disease, a large proportion of patients have limited treatment options and poor prognosis [3]. Therefore, investigating the pathogenesis of iCCA and developing widely applicable treatment strategies are pressing matters that require attention in iCCA research.

CircRNAs are single-stranded, covalently closed RNA molecules that originate from back-splicing of pre-mRNA. The distinct structure of circRNAs affords them an extended half-life and enhanced resistance to RNase R in comparison to linear RNAs. These attributes position circRNAs as promising candidates for diagnostic biomarkers and therapeutic targets. Despite being relatively low in abundance, circRNAs possess a wide array of functions. These include regulating transcription to impact parental gene expression, acting as miRNA sponges to upregulate target mRNAs, recruiting ribosomes for translation, and interacting with proteins to carry out specific functions [4]. Accumulating evidence is now revealing that circRNAs can act as ceRNA by sponging miRNA to relieve the repression of miRNAs for their targets [5]. Hsa_circRNA_0088036 modulates FOXQ1 expression to promote bladder cancer progression by directly interacting with miR-140-3p [6]. CircPGR functions as a ceRNA to sponge miR-301a-5p to accelerate the expression of cell cycle-related genes and breast cancer cell growth [7].

At present, the regulatory mechanism of ceRNA in iCCA remains unclear. In our previous research, hsa_circ_0006834 was the most significantly downregulated among the differential circRNAs between iCCA tissues and adjacent tissues of iCCA patients [15]. And it was predicted that hsa_circ_0006834 can act as miRNA sponge to upregulate the expression of nerve growth factor receptor (NGFR). NGFR, a member of the tumor necrosis factor receptor superfamily, has been revealed to regulate cell autophagy in many tumors [8]. For example, NGFR enhances autophagy induced by 5-FU to inhibit tumor growth in colorectal cancer [9]. Autophagy is a cellular metabolic process that maintains nutrient and energy balance by forming autophagolysosome to degrade cellular contents. It has been proven to be involved in the development of cancer [10]. The attenuation of autophagy facilitates CCA metastasis via a TFEB phosphorylation-dependent manner when PTEN is deficient [11]. ICCA can be inhibited through the induction of autophagy and the accumulation of ROS by aprepitant [12]. Moreover, AMP activated protein kinase (AMPK)-mammalian target of rapamycin (mTOR)-mediated activation of autophagy promotes formation of dormant polyploid giant cancer cells, which is associated with nasopharyngeal carcinoma recurrence [13].

Here, we discovered the hsa_circ_0006834 inhibited the proliferation, migration, and invasion of iCCA cells through regulating autophagy and AMPK-mTOR pathway. Furthermore, it was discovered that hsa_circ_0006834 can act as miRNA sponge to upregulate the expression of NGFR, which is closely associated with cell proliferation and autophagy. This study elucidates the mechanism by which hsa_circ_0006834 promotes AMPK-mTOR pathways, induces autophagy, and ultimately inhibits the

proliferation of iCCA cells via sponging miRNA and upregulating target gene expression. The ceRNA regulatory mechanism of circRNA in iCCA may provide new potential targets for the diagnosis and therapy of iCCA.

## Materials and methods

### 1. Reagents and Plasmids

hsa_circ_0006834 vector plasmids were purchased from Shenggong Bioengineering Shanghai (Shanghai, China).

Si-NC, hsa_circ_0006834-siRNA, siRNA-NGFR and siRNA-AMPK were designed and synthesized by Shenggong Bioengineering Shanghai (Shanghai, China). The siRNA sequences could be found in the Supplementary Table 1.

Primary antibodies were purchased or conducted from Proteintech Group (Wuhan, China), including anti-tubulin, anti-LC3B, anti-p62, anti-NGFR, anti-AMPK, anti-p-AMPK, anti-mTOR and anti-p-mTOR (Ser 2448) antibodies. The goat anti-rabbit IgG, goat anti-mouse IgG as secondary antibodies were purchased from Proteintech Group (Wuhan, China). Autophagy activators (Rapa) or inhibitors (3MA) were purchased from Meilun Biology (Dalian, China).

### 2. Cell culture and transfection

For cell culture, the HCCC-9810 and RBE cells (BeNa Culture Collection, Suzhou, China) were cultured in RPMI-1640 Medium (Biosharp, guangzhou, China) supplemented with 10% fetal bovine serum (FBS) (Vivacell, Shanghai, China) and 1% penicillin/streptomycin (Beyotime, Shanghai, China). Cells were maintained at 37°C in a humidified atmosphere containing 5% $CO_2$.

For cell transfection, a mixed liquid containing liposome (Polyplus Transfection, Strasbourg, France) -plasmid or liposome-siRNA mixture was prepared with serum-free medium and added to the culture plate of HCCC-9810 and RBE cells. After culturing the cells at 37°C for 6–8 h, the mixed solution in the culture plate was replaced with a new serum-containing medium, and cells were collected 48h later.

### 3. Cell counting Kit-8 (CCK-8)

$1 \times 10^3$ iCCA cells/well were seeded into 96-well plate. Then, 10 µL/well CCK-8 solution (Beyotime, Shanghai, China) was added to each well at 0, 24, 48, 72, and 96 h. The optical density (OD) values were measured at 450 nm using a microplate reader [14].

### 4. Transwell assay

For the invasion assay, the transwell chambers (Corning, NY, USA) were covered with Matrigel (BD, San Jose, USA), while the migration assay was performed untreated. The upper chambers received resuspended experimental cells in serum-free culture medium, whereas the lower chambers received complete medium containing 10% serum to stimulate cell migration or invasion. Following culture for 24 h, the bottom chambers were fixed with 4% paraformaldehyde for 0.5 h and dyed with 0.5% crystal violet for 1 h. At last, the stained cells were photographed under a microscope (Leica, Wetzlar, Germany) and Image J was used to count the number of migratory or invasive cells.

### 5. Autophagy inducer and inhibitor were used to treat transfected cells

After transfecting HCCC-9810 and RBE cells with hsa_circ_0006834 vector for 48h, the transfected cells were treated with complete medium containing a final concentration of 3.2µmol/L Rapa or 5 mmol/L 3-MA, and were continued to be cultured in a 5% $CO_2$ incubator at 37°C for 24h.

### 6. RNA extraction, reverse transcription, and Quantitative Real-Time Polymerase Chain Reaction (qRT-PCR)

Total RNA in cells was extracted with TRIzol reagent (Invitrogen, Carlsbad, CA, USA), and the RNA concentration and purity were checked by the enzyme-labeled instrument. RNA was then reverse transcribed into cDNA using the reverse

transcription kit (TransGen Biotech, Beijing, China) according to the instructions provided by the manufacturer. qRT-PCR was carried out with the TransStart Tip Green qPCR SuperMix Kit (TransGen Biotech) in a LightCycler 96 PCR system (Roche, Basel, Switzerland), according to the manufacturer's guidelines. The primers can be found in Supplementary Table 2.

### 7. Western blot

Total protein was isolated by RIPA lysis buffer (Beyotime, Shanghai, China) with 1% PMSF (Beyotime, Shanghai, China) and quantified by BCA Protein Assay Kit (Beyotime, Shanghai, China). Thirty micrograms of protein were separated by SDS-PAGE (Beyotime, Shanghai, China) and transferred to polyvinylidene di-fluoride (PVDF) membranes (MerckMillipore, shanghai, China). Proteins were incubated with primary antibody detecting p62 (1:1000), NGFR (1:1000), p-mTOR (1:5000), mTOR (1:5000), p-AMPK (1:1000), AMPK (1:1000), and LC3B (1:1000), β-tubulin (1:5000) was used as a control. The anti-rabbit and anti-mice secondary antibodies were then applied (1:5000). Finally, enhanced chemiluminescence was utilized to observe immunoreactive proteins.

### 8. Dual-luciferase reporter assay

The synthetic interaction and mutation sequence (Supplementary Table 3) were inserted between SacI/XbaI in pmirGLO vector to construct the dual-luciferase reporter gene vector and mutant (Sangon Biotech, shanghai, China). In six-well plates, RBE and HCCC-9810 cells were cultured to co-transfected with either wild type or mutant luciferase reporter vector (2 μg) and either mimic miRNAs or negative control (NC) (2 μg). After 48 h, luciferase activity was measured and normalized to the activity of Rluc.

### 9. RNA Pull-down assay

Hsa_circ_0006834 biotinylated probe (5'-ATCTCACTGATCTCTCCCCAGC-3') and the mutated biotinylated probe that disrupts the predicted miR-637 binding site (5'- ATCTCACTGATCTCTAAAAAGAC −3') were synthesized by Shenggong Bioengineering Shanghai (Shanghai, China). Hsa_circ_0006834 was labeled with biotin and incubated with streptavidin beads (Thermo, CA, USA) at 4 °C overnight. The mixture was then centrifuged at 3000 rpm for 1 min and washed with Wash buffer three times. A total of $2 \times 10^7$ cells were lysed. The bead-biotin complex was added to the lysates and incubated at room temperature (RT) for 1 h. After washing with Wash buffer II, the RNA that was bound to the bead was captured and extracted with Trizol for the subsequent qPCR assay.

### 10. Statistical analysis

All results in this study were analyzed using SPSS 19.0 statistical software; GraphPad Prism 6.0 software (GraphPad Software, Inc.) was used to generate graphs. Values are expressed as mean ± standard deviation from at least 3 experiments. Statistical analyses were performed using One-way or Two-way ANOVA test to determine statistical significance between groups. $P < 0.05$ was considered a statistically significant difference.

## Results

### 1. Hsa_circ_0006834 attenuates proliferation and migration of iCCA cells

Hsa_circ_0006834 was the most significantly downregulated among the differential circRNAs between iCCA tissues and adjacent tissues of iCCA patients [15]. To investigate the effects of hsa_circ_0006834 on iCCA, a hsa_circ_0006834 vector was constructed and transfected into HCCC-9810 and RBE cells. In the CCK-8 assay, the overexpression of hsa_circ_0006834 significantly inhibited the proliferation of HCCC-9810 and RBE cells (Fig 1A, B). In the transwell assay, hsa_circ_0006834 overexpression inhibited the migration and invasion of HCCC-9810 and RBE cells (Fig 1C–F). These results suggested that hsa_circ_0006834 attenuates the proliferation, migration and invasion of iCCA cells.

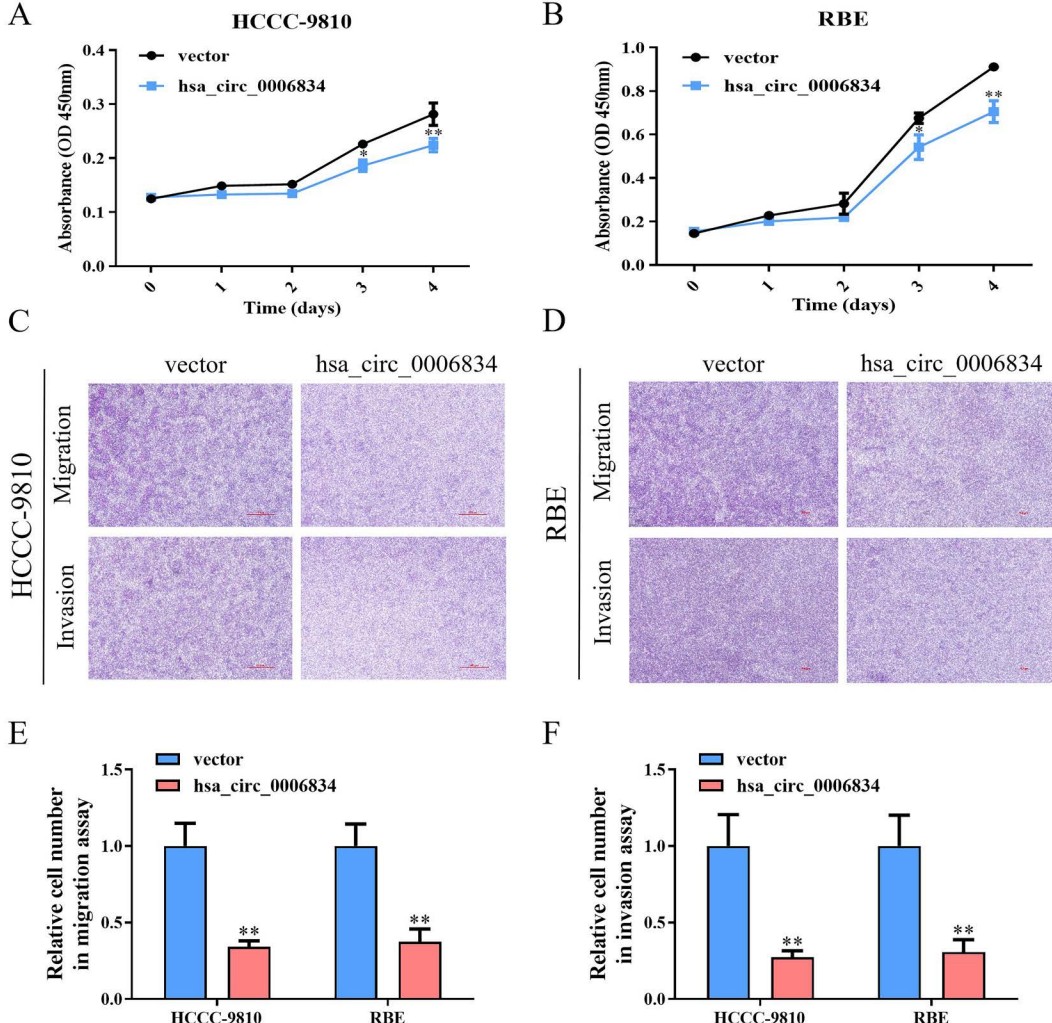

**Fig 1. Hsa_circ_0006834 inhibited iCCA cells proliferation, migration and invasion.** (A–B) CCK-8 assay was conducted to evaluate the cell viability of iCCA cells that had been transfected with hsa_circ_0006834 plasmids. (C–F) Assessment of the effect of hsa_circ_0006834 on iCCA cells migration and invasion. Transwell assay was established for estimating cell migration and invasion (the resulting statistical graphs are shown in Fig 1E–F). Error bars represent three independent experiments. *, **, *** indicates significant differences compared with the control group at a p value < 0.05, < 0.01, < 0.001, respectively.

## 2. Hsa_circ_0006834 inhibits the proliferation of iCCA cells via autophagy

Studies have shown that cell proliferation and autophagy are closely related [16]. Therefore, we observed the effects of hsa_circ_0006834 on autophagy. Hsa_circ_0006834 vector or hsa_circ_0006834-siRNAs were transfected into HCCC-9810 and RBE respectively. And the expression levels of autophagy-related genes (ATG3, ATG12, LC3B, BECLIN1) and proteins were detected by qPCR and Western blot. The results showed that the expression levels of autophagy-related genes and LC3BII/LC3BI ratio were significantly increased, the P62 protein level was decreased in hsa_circ_0006834 overexpressed HCCC-9810 and RBE (Fig 2A–F). And the changes were contrary in the hsa_circ_0006834 knockdown HCCC-9810 and RBE cells compared with si-NC group (Fig 2A–F). These results indicated that hsa_circ_0006834 promoted autophagy and the autophagy flow was smooth.

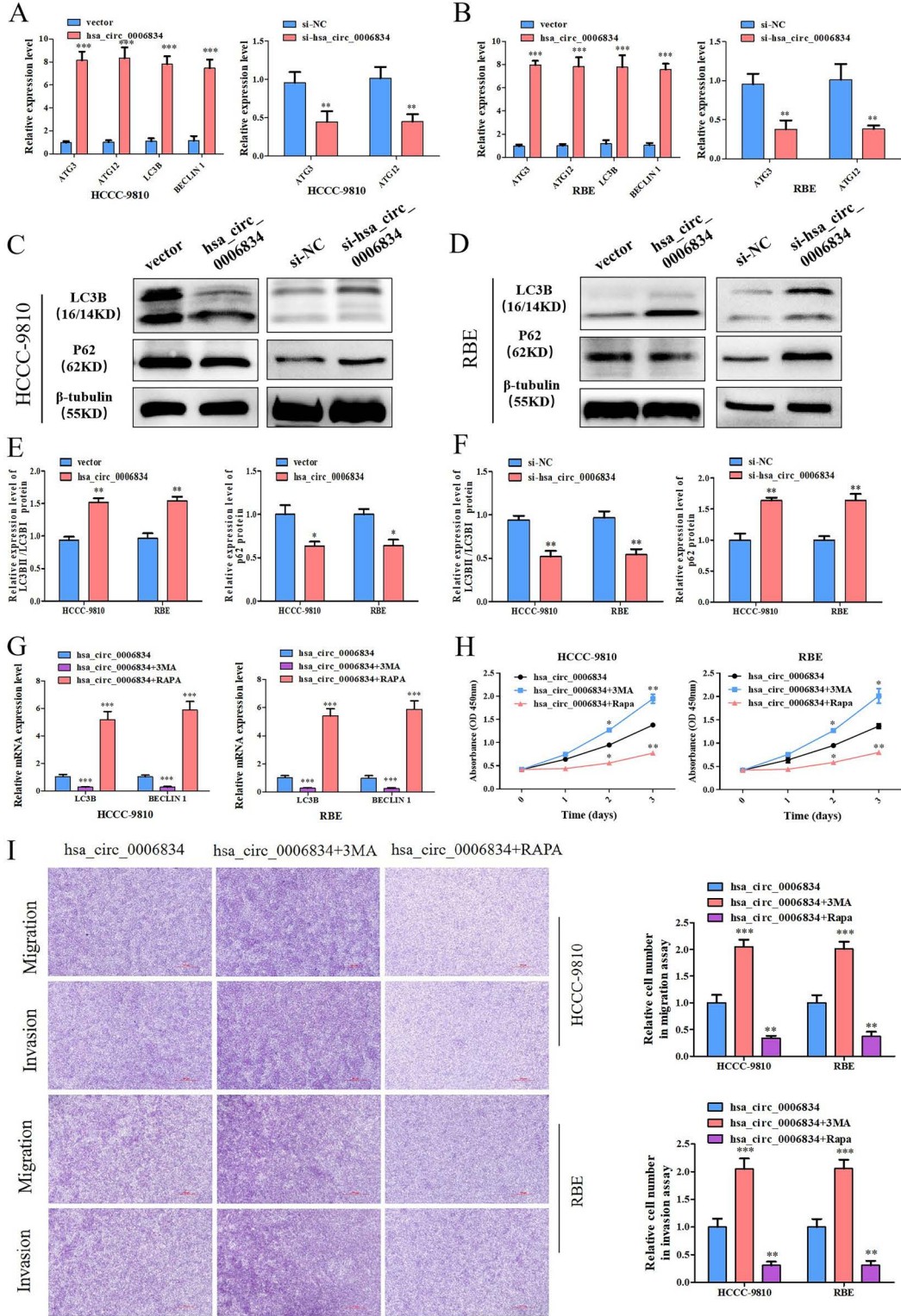

**Fig 2. Hsa_circ_0006834 inhibited the proliferation, migration, and invasion of iCCA cells by inducing autophagy.** (A–B) The effect of hsa_circ_0006834 on autophagy related genes expression was detected by qPCR. (C-F) The effect of hsa_circ_0006834 on autophagy related proteins expression was detected by Western blot (the resulting statistical graphs are shown in Fig 2E–F). (G) The effects of autophagy activator RAPA and

autophagy inhibitor 3MA on the expression of autophagy related genes in hsa_circ_0006834 overexpressing cells were detected by qPCR. (H) CCK-8 assay was performed to detect the effects of RAPA and 3MA on the proliferation of hsa_circ_0006834 overexpressing cells. (I) Transwell assay was established for estimating the effects of RAPA and 3MA on the migration and invasion of hsa_circ_0006834 overexpressing cells. The right images are the statistical graphs of the left figure. Error bars represent three independent experiments. *, **, *** indicate significant differences compared with the control group at p values < 0.05, < 0.01, < 0.001, respectively.

To further clarify whether hsa_circ_0006834 affects cell proliferation and migration through autophagy [17], we treated hsa_circ_0006834 overexpressed HCCC-9810 and RBE cells with 3MA (the autophagy inhibitor) or RAPA (the autophagy activator) respectively. MTT assay showed there was no cytotoxicity with 5 mM 3MA-treated or 3.2 μM RAPA-treated cells (Date not shown). Compared to the hsa_circ_0006834 group, the expression levels of autophagy genes were significantly inhibited (activated) in the hsa_circ_0006834 and 3MA (RAPA) co-treatment groups (Fig 2G). In CCK-8 assay, the proliferation of cells was significantly increased treated with 3MA, while reduced treated with RAPA (Fig 2H). In the transwell assay, the migration and invasion of cells treated with 3MA were significantly improved, while inhibited treated with RAPA (Fig 2I). These results suggested that hsa_circ_0006834 inhibits the proliferation, migration and invasion of iCCA cells through autophagy, and there is a negative correlation.

### 3. The regulatory network of hsa_circ_0006834-has-miR-637-NGFR

The potential target miRNAs of hsa_circ_0006834 were predicted for mechanism by miRBase, and 13 miRNAs, including miR-637, were binded to hsa_circ_0006834 (Fig 3A). RNA pull-down was performed with biotin labeled probe targeting hsa_circ_0006834 and mutated probe, the expression levels of miR-637 was upregulated in the precipitated complex extracted for qPCR, and the mutated probe ablated miR-637 binding (Fig 3B). The luciferase activity was significantly reduced after co-transfecting hsa_circ_0006834 and miR-637 mimic in RBE cells, which was not observed in the hsa_circ_0006834 mutant group, according to a dual-luciferase reporter experiment (Fig 3C, D). Furthermore, miR-637 expression was obviously decreased in HCCC-9810 and RBE cells treated with the hsa_circ_0006834 overexpression plasmid (Fig 3E). MiR-637 expression was higher in iCCA cells compared to HIBEC, and it was adversely related to hsa_circ_0006834 expression (Fig 3F). These studies found that miR-637 was a downstream target of hsa_circ_0006834.

NGFR was a potential target of miR-637, which was unveiled by TargetScan (Fig 3G). In dual-luciferase reporter assay, co-transfection of miR-637 mimic and NGFR reporter plasmid remarkably reduced the luciferase intensity of RBE cells (Fig 3H). In iCCA cells and tumor tissues, NGFR mRNA level was considerably lower than in HIBEC and normal tissues (Fig 3I).

In further experiments, hsa_circ_0006834 vectors were transfected in HCCC-9810 and RBE, and the expression levels of NGFR mRNA and protein respectively detected by qPCR and Western blot. The results showed that the overexpression of hsa_circ_0006834 significantly promoted the expression of NGFR gene and protein (Fig 3J–L). The above study confirmed that hsa_circ_0006834 activates the expression of the downstream target gene NGFR by adsorbing miR-637.

### 4. Hsa_circ_0006834 regulates autophagy and proliferation of iCCA cells through NGFR

In order to explore the role of NGFR in hsa_circ_0006834 regulation of autophagy and proliferation, si-NGFR, a small interfering RNA of NGFR, was synthesized and transfected into hsa_circ_0006834 overexpression HCCC-9810 and RBE. The interference efficiency of NGFR was first detected (Fig 4A, C, D). Subsequently, the qPCR results showed the expression levels of autophagy genes LC3 and BECLIN1 were significantly inhibited (Fig 4B), and the Western blot results revealed the ratio of LC3BII/LC3BI was decreased and the P62 was increased (Fig 4C, D). Furthermore, the CCK-8 proliferation assay and transwell assay displayed the proliferation, migration and invasion were upregulated in hsa_circ_0006834 and si-NGFR co-transfected group (Fig 4E–G). These studies demonstrated that hsa_circ_0006834 promotes autophagy and inhibits proliferation, migration and invasion through NGFR.

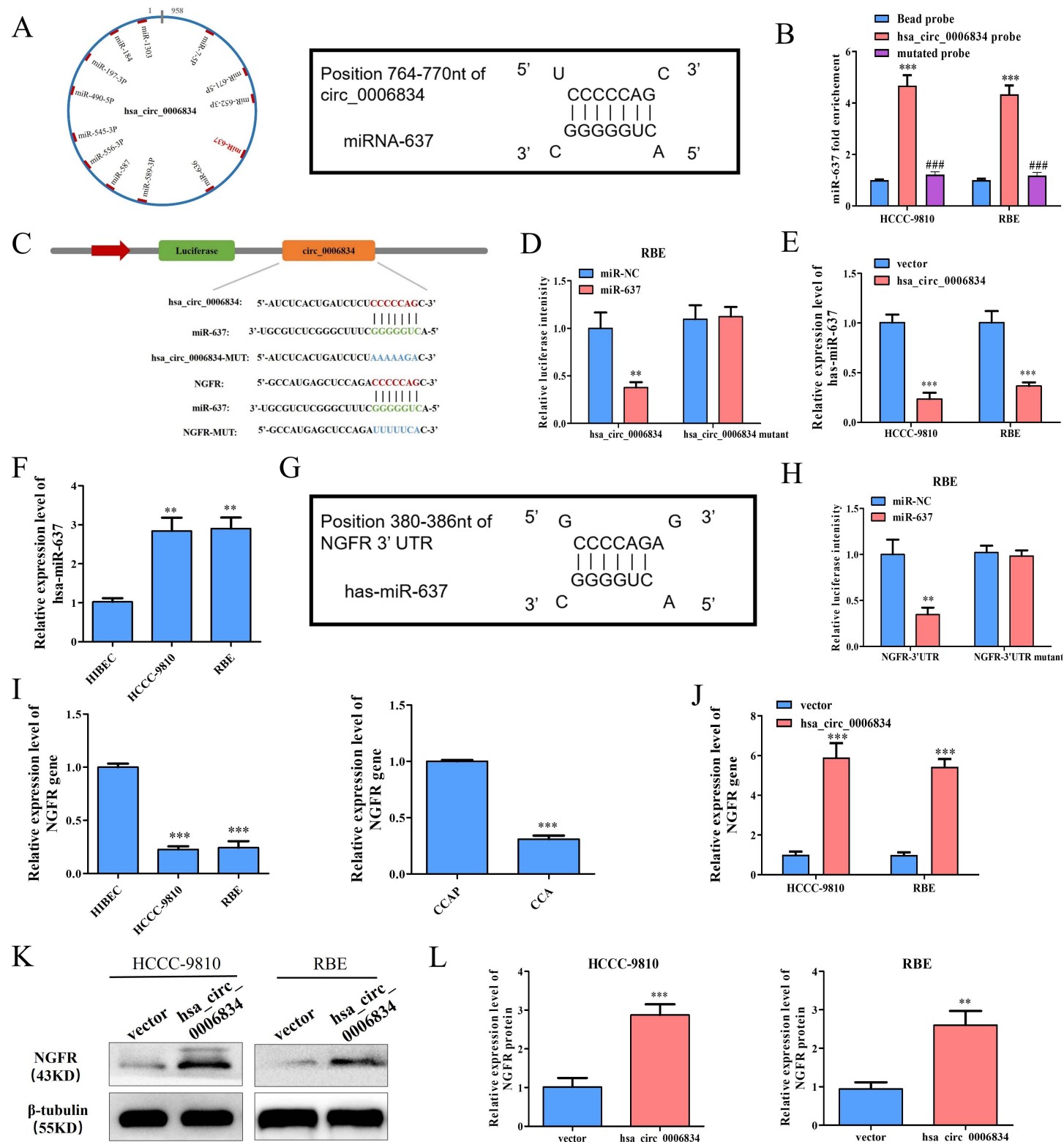

Fig 3. The regulatory network of hsa_circ_0006834-has-miR-637-NGFR. (A) The prediction of miRNA related to hsa_circ_0006834. Schematic representation of the predicted binding sites of miRNAs related to hsa_circ_0006834 is shown on the left. The binding sets between miRNA-637 and hsa_circ_0006834 are presented on the right. (B) The biotin-labeled probe targeting hsa_circ_0006834 or mutated probe

was used for pull-down experiment, and RNA in the pull-down complex was extracted for qRT-PCR to detect the expression of miR-637. (C) Schematic representation of hsa_circ_0006834 or NGFR wild-type (WT) and mutant (Mut) luciferase reporter vectors and miR-637 binding sites. (D) The relative luciferase activities were detected after co-transfection with miR-NCs or miR-637 mimics and hsa_circ_0006834 WT and Mut luciferase reporter vectors in RBE cells. (E) The effect of hsa_circ_0006834 on the expression level of has-miR-637 was detected by qPCR. (F) The relative expression level of hsa-miR-637 in iCCA cells was detected by qPCR. (G) The binding sets between miRNA-637 and NGFR-3'UTR are presented. (H) The relative luciferase activities were detected after co-transfection with miR-NCs or miR-637 mimics and NGFR-3'UTR WT and Mut luciferase reporter vectors in RBE cells. (I) The relative expression level of NGFR gene in iCCA cells and tissues were detected by qPCR respectively. (J) The effect of hsa_circ_0006834 on the expression level of NGFR gene was detected by qPCR. (K-L) Western blot was conducted to explore the effect of hsa_circ_0006834 on the expression level of NGFR protein (the resulting statistical graphs are shown in Fig 3L). Error bars represent three independent experiments. *, **, *** indicate significant differences compared with the control group at p values < 0.05, < 0.01, < 0.001, respectively. ### indicates significant differences compared with the hsa_circ_0006834 probe group at p values <0.001.

## 5. Hsa_circ_0006834 activates the AMPK-mTOR pathway through NGFR

NGFR is associated with the AMPK-mTOR pathway [8]. To confirm that hsa_circ_0006834 regulates the AMPK-mTOR pathway through NGFR, we examined the effect of hsa_circ_0006834 on the AMPK-mTOR pathway at first. The hsa_circ_0006834 vector was transfected into HCCC-9810 and RBE. In Western Blot assay, the ratio of p-AMPK/AMPK protein was upregulated and the ratio of p-mTOR/mTOR protein was downregulated (Fig 5A, B). The results showed that hsa_circ_0006834 could promote AMPK phosphorylation and inhibit mTOR phosphorylation.

Then, we explored the role of NGFR in hsa_circ_0006834 regulation of the AMPK-mTOR pathway. Hsa_circ_0006834 vector with si-NC or si-NGFR were co-transfected in HCCC-9810 and RBE respectively. The p-AMPK/AMPK protein ratio was downregulated and the p-mTOR/mTOR protein ratio was upregulated by Western Blot in hsa_circ_0006834 and si-NGFR co-transfected group (Fig 5C, D). These results indicated that hsa_circ_0006834 could activate the AMPK-mTOR pathway through NGFR.

## 6. Hsa_circ_0006834 induces autophagy and inhibits the proliferation of iCCA cells through AMPK

To investigate the manner in which the AMPK-mTOR pathway regulates autophagy and proliferation via hsa_circ_0006834, HCCC-9810 and RBE cells were co-transfected with the hsa_circ_0006834 vector in combination with either si-NC or si-AMPK. The successful knockdown of AMPK was verified by Western Blot (Fig 6A, B). In the hsa_circ_0006834 and si-AMPK co-transfected group, the LC3BII/LC3BI ratio was downregulated and the P62 protein level was upregulated (Fig 6A, C), indicating that hsa_circ_0006834 induces autophagy through AMPK. Additionally, the increased p-mTOR/mTOR protein ratio revealed that mTOR functions as a downstream signal of AMPK (Fig 6A, D). Cell proliferation ability, migration, and invasion were all enhanced following si-AMPK interference, as observed in CCK-8 and transwell assays (Fig 6E-G). These results indicated that hsa_circ_0006834 could attenuate cell proliferation, migration and invasion through the AMPK-mTOR pathway. According to the aforementioned studies, hsa_circ_0006834 could promote NGFR expression, activate the AMPK-mTOR pathway, induce autophagy and inhibit iCCA cell proliferation, migration and invasion.

## Discussion

CircRNAs represent a distinct class of non-coding RNAs with covalently closed loop structures, whose outstanding stability makes them ideal biomarkers for the diagnosis and prognosis of iCCA [18]. Mounting evidence indicates that the dysregulation of circRNA expression is associated with iCCA pathogenesis and development. For example, circNFIB, whose loss is highly associated with aggressive characteristics, competitively interacts with MEK1, suppressing MEK1/ERK signaling and iCCA metastasis [19]. An upregulated circRNA cPKM can promote STMN1 expression to enhance the proliferation and metastasis of iCCA cells by sponging miR-199a-5p [20].

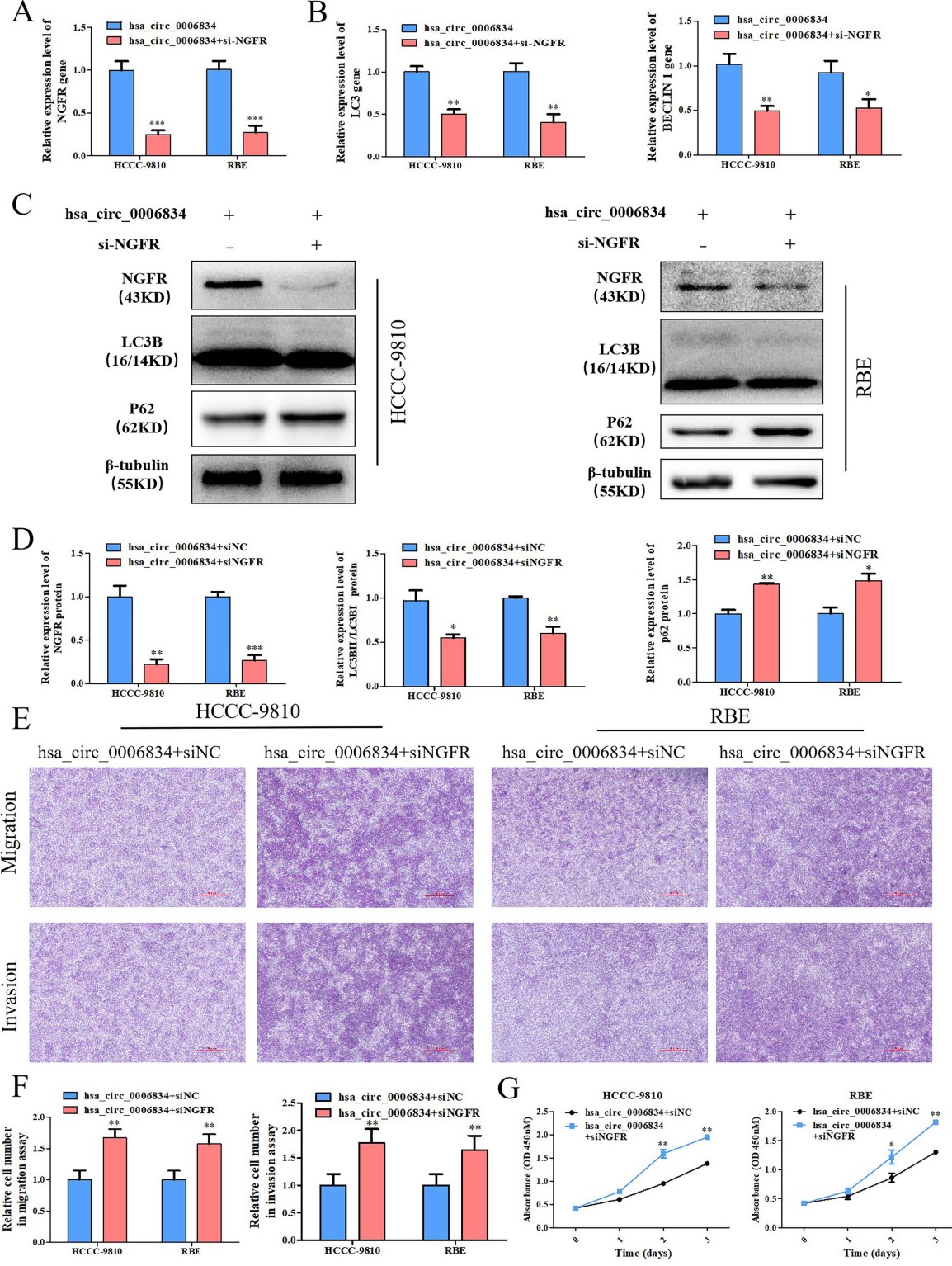

**Fig 4. Hsa_circ_0006834 promoted autophagy and inhibited proliferation, migration and invasion through NGFR.** (A) The interference efficiency of si-NGFR was detected by qPCR. (B) The effect of si-NGFR treatment on autophagy related genes in hsa_circ_0006834 transfected cells was detected by qPCR. (C-D) Western blot was conducted to explore the effect of si-NGFR treatment on NGFR protein and autophagy related proteins

in hsa_circ_0006834 transfected cells. The resulting statistical graphs are shown in Fig 3D. (E-F) Transwell assay was established for estimating cell migration and invasion after si-NGFR treatment in hsa_circ_0006834 transfected cells. The resulting statistical graphs are shown in Fig 4F. (G) CCK-8 assay was performed to examine cell viability of iCCA cells transfected with hsa_circ_0006834 and si-NGFR or si-NC respectively. Error bars represent three independent experiments. *, **, *** indicate significant differences compared with the control group at p values < 0.05, < 0.01, < 0.001, respectively.

CircRNA hsa_circ_0006834, which is markedly downregulated in iCCA cells and tissue, has been proved to inhibit the proliferation, migration, invasion of iCCA cells in our previous studies. Due to the diversity of circRNAs functions, the various mechanism of hsa_circ_0006834 on iCCA needs further exploration. Here, we investigated the role of hsa_circ_0006834 in iCCA from the perspective of ceRNA regulatory mechanism, revealing its potential functions.

The circRNA-miRNA-mRNA axis is a widely occurring pattern of gene expression regulation [21]. AKT3 and mTOR relative mRNA levels may be upregulated in RBE cells as a result of circ-CDR1as binding to and accelerating miR-641 degradation. This would activate the AKT3/mTOR pathway and encourage the growth of CCA [22]. Here, we discovered that hsa_circ_0006834 was mainly localized in the cytoplasm and directly interacted with miR-637, serving as a sponge for the latter and preventing it from controlling the expression of NGFR.

NGFR, also known as p75NTR or CD271, influences numerous biological processes, including cell division and death, energy expenditure, scar formation, and hypoxia response [23]. Additional research has shown that in many cancers, NGFR controls cell proliferation, invasion, metastasis, autophagy, and apoptosis [9,23]. Melanoma-derived stem cells release NGFR, which promotes metastasis and lymph node pre-metastatic niche development [24]. Additionally, NGFR may be involved in the perineural invasion of pancreatic cancer cells, mediating the chemoattraction of cancer cells for neural tissues [25]. Here, we discovered that the low expression of NGFR induced the development of iCCA and NGFR was associated with autophagy and proliferation. Studies have shown that NGFR regulates autophagy by disrupting the binding of Bcl-2 and Beclin-1 proteins [9]. In this study, we found that hsa_circ_0006834 could promote autophagy and inhibit iCCA cell proliferation, migration, and invasion through NGFR.

AMPK and mTOR are major positive and negative regulators of autophagy, respectively [26]. Based on a new study, resveratrol stimulates the NGFR downstream pathway AMPK-mTOR to induce autophagy in non-small-cell lung cancer cells [8]. In osteosarcoma, TRIM22 inhibits tumor progression via ROS/AMPK/mTOR-mediated autophagy activation [27]. In this study, hsa_circ_0006834 was able to promote the phosphorylation of AMPK and restrain the phosphorylation of mTOR by means of NGFR. Furthermore, we verified that hsa_circ_0006834 is associated with AMPK-mTOR pathway activation, which in turn suppresses proliferation. While our data support a role for this pathway, other mechanisms may also contribute, as highlighted by the circRNA's potential interactions with other molecules.

We acknowledge limitations in our study. The reliance on *in vitro* cell line data may not fully recapitulate in vivo tumor complexity [28]. Hsa_circ_0006834 might exert functions beyond the investigated axis, such as binding to other miRNAs or RBPs. In future research, we explore the possibility of hsa_circ_0006834 functioning by translating small peptides. These gaps highlight the need for *in vivo* models and comprehensive regulatory network analyses in future studies.

In the landscape of tumor diagnosis and therapy, several innovative approaches have emerged as pivotal strategies against breast cancer and other malignancies. Targeted imaging of tumor-associated macrophages enables precise visualization of the tumor microenvironment [28,29], while copper-based nanomaterials integrate diagnostic imaging with therapeutic functionality [30]. Intermittent perturbation of F-actin cytoskeleton has shown promise in inhibiting metastatic progression [31], and Ayurvedic botanicals targeting the progesterone receptor offer novel endocrine modulation strategies [32]. Concurrently, traditional Chinese medicines have demonstrated clinical benefits in primary liver cancer management [33].

For predictive oncology, artificial intelligence-driven facial feature analysis represents a non-invasive risk assessment tool [34], while miR-29a-3p has illuminated the molecular basis of exercise-induced anti-cancer immunity [35]. Collectively, these advancements underscore the importance of multi-faceted strategies in oncology. Notably, the ceRNA regulatory

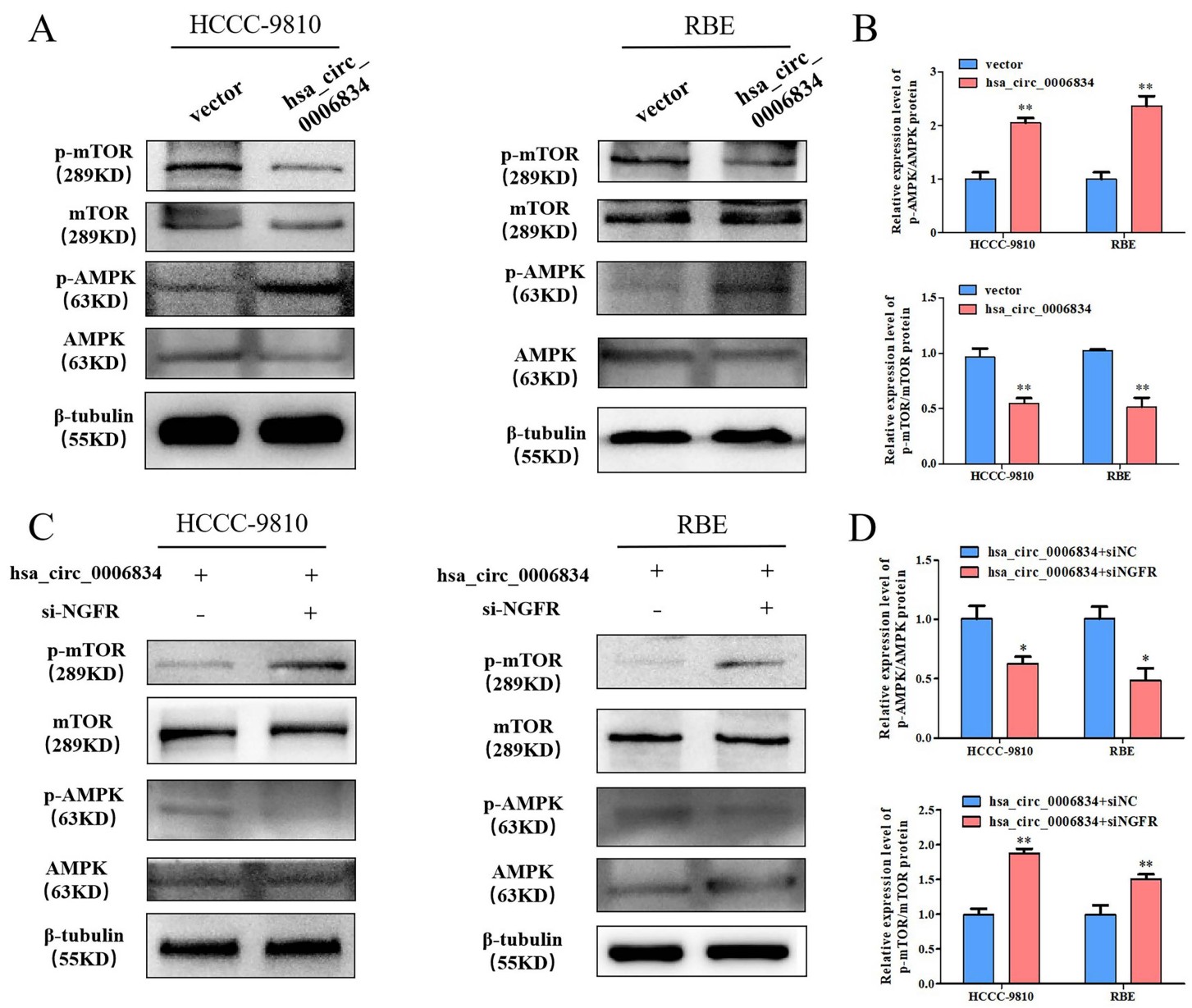

**Fig 5. Hsa_circ_0006834 activated the AMPK-mTOR pathway through NGFR.** (A–B) Western blot assay was conducted to analyze the effect of hsa_circ_0006834 on the expression level of AMPK-mTOR protein. The resulting statistical graphs are shown in Fig 5B. (C–D) The effect of si-NGFR treatment on the expression level of AMPK-mTOR protein in hsa_circ_0006834 transfected cells was detected by Western blot. The resulting statistical graphs are shown in Fig 5D. Error bars represent three independent experiments. *, **, *** indicate significant differences compared with the control group at p values < 0.05, < 0.01, < 0.001, respectively.

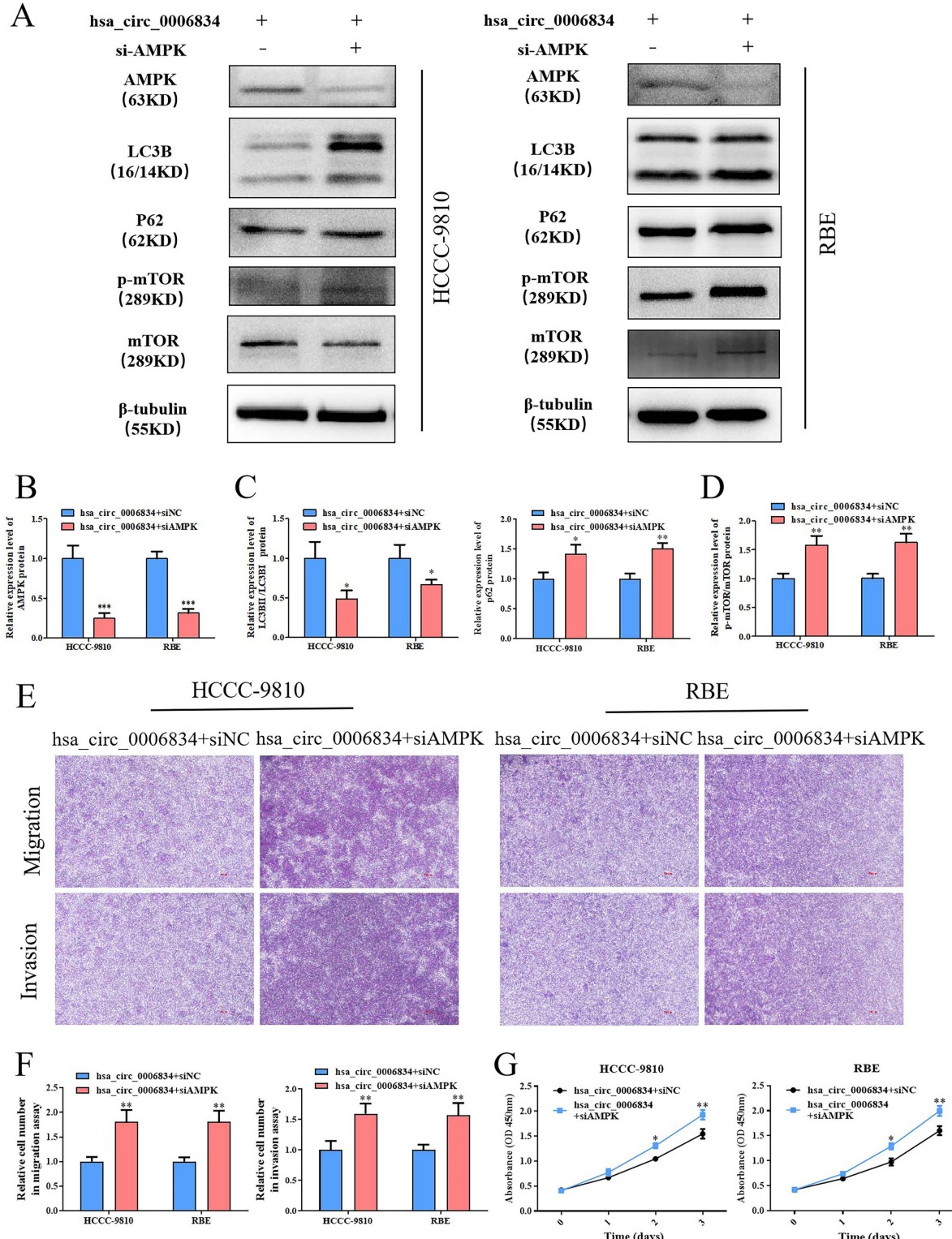

**Fig 6. Hsa_circ_0006834 induced autophagy and inhibited the proliferation, migration and invasion of iCCA cells through AMPK.** (A–D) Western blot was conducted to assess the protein expression levels of AMPK, LC3B, P62, p-mTOR and mTOR for three specific objectives sequentially: to validate the knockdown of AMPK (Fig 6A and B), to investigate the effect of si-AMPK interference on autophagy-related proteins expression in

hsa_circ_0006834 transfected cells (Fig 6A and C), and to examine the connection between mTOR and AMPK (Fig 6A and D). (E-F) Transwell assay was utilized to estimate cell migration and invasion subsequent to si-AMPK treatment in hsa_circ_0006834 transfected cells, with the resulting statistical graphs depicted in Fig 6F. (G) CCK-8 assay was applied to assess cell viability of iCCA cells transfected with hsa_circ_0006834 and si-AMPK or si-NC, respectively. Error bars represent three independent experiments. *, **, *** indicate significant differences compared with the control group at p values < 0.05, < 0.01, < 0.001, respectively.

mechanism of circRNAs in intrahepatic cholangiocarcinoma (iCCA) emerges as a promising frontier, potentially unveiling new diagnostic biomarkers and therapeutic targets for precision oncology.

## Supporting information

**S1 Table. The sequences of siRNAs. The left column show ed the gene names of siRNAs. The right column showed the corresponding sense and antisense sequences of siRNAs.**
(JPG)

**S2 Table. The sequences of primers. The left column showed the gene names. The middle column showed the corresponding Primer names. The right column showed the corresponding sequences of Primers.**
(JPG)

**S3 Table. The synthetic interaction and mutation sequence. The left column showed the names of the dual-luciferase reporter gene vector and mutant. The right column showed the corresponding sequences.**
(JPG)

**S4 File.**
(PDF)

## Author contributions

**Conceptualization:** Ji Wang, Xinyue Qi, Yue Wang, Yunshan Zhang.

**Data curation:** Xinyue Qi, Wangqian Zhao, Ziyan Hao, Kehan Wang, Yuting Li.

**Formal analysis:** Xinyue Qi, Wangqian Zhao, Ziyan Hao, Kehan Wang, Yuting Li.

**Funding acquisition:** Ji Wang, Yunshan Zhang.

**Investigation:** Xinyue Qi, Wangqian Zhao, Ziyan Hao, Kehan Wang, Yuting Li.

**Project administration:** Yue Wang.

**Supervision:** Yunshan Zhang.

**Writing – original draft:** Yunshan Zhang.

**Writing – review & editing:** Xinyue Qi, Yue Wang, Yunshan Zhang.

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
