## [Decision Letter · Decision Letter 0]

13 Jun 2025

PONE-D-25-12011Hsa_circ_0006834 represses intrahepatic cholangiocarcinoma proliferation through activating AMPK-mTOR pathway and autophagy via has-miR-637-NGFR networkPLOS ONE

Dear Dr. Zhang,

Thank you for submitting your manuscript to PLOS ONE. After careful consideration, we feel that it has merit but does not fully meet PLOS ONE’s publication criteria as it currently stands. Therefore, we invite you to submit a revised version of the manuscript that addresses the points raised during the review process.

**ACADEMIC EDITOR: ** Thank you for submitting your manuscript to the Journal and as voucan see that the reviewer think your manuscript is interesting and provide valuable comments for your reference. Please submit the revised manuscript ASAP and also include a rebuttal that would clearly list all the responses to the reviewer's comments.

We look forward to receiving your revised manuscript.

Kind regards,

Zhiwen Luo

Academic Editor

PLOS ONE

“This work was supported by the National Natural Science Foundation of China (Grant No. 32200110), the Natural Science Foundation of the Jiangsu Higher Education Institutions of China (22KJB230002), Provincial-level talent program for National center of technologyinnovation for biopharmaceuticals (NCTIB2024JS0101). The funders had no role in study design, data collection, analysis, decision to publish, or manuscript preparation.”

5. Please upload a new copy of Figures as the detail are not clear. Please follow the link for more information: https://blogs.plos.org/plos/2019/06/looking-good-tips-for-creating-your-plos-figures-graphics/"
https://blogs.plos.org/plos/2019/06/looking-good-tips-for-creating-your-plos-figures-graphics/

Additional Editor Comments:

Thank you for submitting your manuscript to the Journal and as voucan see that the reviewer think your manuscript is interesting and provide valuable comments for your reference. Please submit the revised manuscript ASAP and also include a rebuttal that would clearly list all the responses to the reviewer's comments.

Reviewers' comments:

Reviewer's Responses to Questions

**Comments to the Author**

1. Is the manuscript technically sound, and do the data support the conclusions?

Reviewer #1: Yes

Reviewer #2: Yes

2. Has the statistical analysis been performed appropriately and rigorously? 

Reviewer #1: Yes

Reviewer #2: Yes

3. Have the authors made all data underlying the findings in their manuscript fully available?

Reviewer #1: Yes

Reviewer #2: Yes

4. Is the manuscript presented in an intelligible fashion and written in standard English?

Reviewer #1: Yes

Reviewer #2: Yes

5. Review Comments to the Author

Reviewer #1: Dear Authors,

I have completed the review of your manuscript titled "Hsa_circ_0006834 represses intrahepatic cholangiocarcinoma proliferation through activating AMPK-mTOR pathway and autophagy via has-miR-637-NGFR network". I'm delighted to inform you that I recommend your manuscript for acceptance.

Your study has impressively explored the role and mechanism of hsa_circ_0006834 in intrahepatic cholangiocarcinoma (iCCA). The overall experimental design is well-structured, the data analysis is thorough, and the biological interpretations are both sound and convincing. You've done a remarkable job in elucidating the molecular network and demonstrating its impact on the key biological processes related to iCCA, thereby adding valuable knowledge to the field.

However, there is a minor aspect that I think could benefit from some improvement. Specifically, I noticed that the scale bars on the cell staining photos in your manuscript are not as clear as they could be. This might pose an obstacle for readers when they are trying to accurately understand and interpret the results shown in these photos. To enhance the clarity and readability of the visual data presentation, I suggest that you replace the existing scale bars. This is a relatively straightforward adjustment but can significantly improve the overall quality of how your visual data is presented.

Once this small modification is made, your manuscript will be in excellent shape for publication. I'm looking forward to seeing it published in due course.

Best regards,

Lei

Reviewer #2: This manuscript investigates the role of a circular RNA, hsa_circ_0006834, in intrahepatic cholangiocarcinoma (iCCA). The study combines in vitro experiments using iCCA cell lines to propose a mechanism where hsa_circ_0006834 functions as a competing endogenous RNA (ceRNA). The authors suggest that hsa_circ_0006834 sponges has-miR-637, leading to the upregulation of its target, nerve growth factor receptor (NGFR). This, in turn, is proposed to activate the AMPK-mTOR signaling pathway, induce autophagy, and ultimately suppress iCCA cell proliferation, migration, and invasion. While the manuscript presents a cohesive and interesting narrative with a logical series of experiments, several significant aspects require further experimental validation and clarification to strengthen the conclusions. The manuscript also needs improvements in its structure, logical flow, and linguistic precision.

Major revisions are required:

To more robustly establish the proposed ceRNA mechanism, have the authors considered performing critical rescue experiments? For instance, could the inhibitory effects of hsa_circ_0006834 overexpression on cell proliferation, migration, and autophagy be reversed by the co-transfection of a has-miR-637 mimic? This would provide stronger evidence that the circRNA's function is mediated primarily through sponging this specific miRNA.

The RNA pull-down assay demonstrated an enrichment of miR-637 with the biotinylated hsa_circ_0006834 probe. To confirm the specificity of this interaction, it would be beneficial to include a control using a mutated biotinylated probe that disrupts the predicted miR-637 binding site. This would demonstrate that the interaction is sequence-specific.

While the data show that silencing NGFR reverses the effects of hsa_circ_0006834 overexpression, the mechanism linking NGFR to AMPK activation is not directly elucidated. The manuscript states NGFR is associated with the pathway, citing another study. Can the authors provide more direct mechanistic insight in their specific iCCA cell model? For example, does NGFR interact with AMPK or an upstream kinase?

The study convincingly shows that silencing AMPK reverses the pro-autophagic and anti-proliferative effects of hsa_circ_0006834. However, in these AMPK knockdown experiments, has the phosphorylation status of downstream mTOR targets, such as p70S6K or 4E-BP1, been assessed to further confirm the modulation of the entire AMPK-mTOR axis?

The manuscript relies heavily on two iCCA cell lines (HCCC-9810 and RBE). While this is a standard approach, have the authors validated their key findings, such as the expression levels of hsa_circ_0006834, miR-637, and NGFR, in a cohort of human iCCA patient tissues versus adjacent non-tumorous tissues? This would significantly enhance the clinical relevance of the findings. The manuscript mentions that NGFR was lower in tumor tissues but the data presentation is limited.

The introduction is adequate but could be more focused. It broadly discusses autophagy, AMPK-mTOR, and circRNAs. It would be more impactful to streamline this section to build a stronger, more direct rationale for investigating the specific hsa_circ_0006834/miR-637/NGFR axis in the context of iCCA from the outset.

The references cited in this article are not sufficient, and there is a lack of in-depth comparative discussion. Background and methodology also require further literature support. Some related research should be cited:

10.15212/bioi-2022-0010

10.15212/bioi-2024-0013

10.15212/bioi-2024-0012

10.15212/bioi-2024-0066

10.34133/research.0080

10.34133/research.0033

10.5847/wjem.j.1920-8642.2022.015

10.1016/j.celrep.2024.114697

10.53388/tmr20220207263

10.53388/tmr20211227255

There are several typographical errors and inconsistencies within the figure legends that detract from the manuscript's quality. For instance, the legend for Figure 1 refers to "Fig. 1F-G" while the figure itself only has panels up to F. Similarly, the legend for Figure 2 refers back to statistical graphs in "Fig. 1E-F", and Figure 3's legend refers to "Fig. 2L". The entire manuscript requires careful proofreading for such errors.

The terminology for key reagents is inconsistent. For example, the circRNA is referred to as "hsa_circ_0006834", "circRNA_0006834", and "hsa circ 0006834" throughout the text and figures. Please select a single, consistent nomenclature and apply it throughout the manuscript for clarity.

The Discussion section effectively summarizes the results but would be strengthened by a more critical analysis of the findings. Please include a subsection discussing the limitations of the study, such as the reliance on in-vitro data and the possibility that this circRNA may have other functions or interact with other molecules beyond the specific axis investigated. This would provide a more balanced perspective.

The conclusion that hsa_circ_0006834 represses iCCA proliferation through activating the AMPK-mTOR pathway is a strong claim. While the data link these events, causality is implied but not definitively proven to be the sole mechanism. The authors should consider moderating this language to state that hsa_circ_0006834 is associated with or contributes to the activation of this pathway, which in turn suppresses proliferation. This would more accurately reflect the nature of the evidence provided.

6. PLOS authors have the option to publish the peer review history of their article (what does this mean? ). If published, this will include your full peer review and any attached files.

**Do you want your identity to be public for this peer review?** For information about this choice, including consent withdrawal, please see our Privacy Policy .

Reviewer #1: No

Reviewer #2: No

---

## [Author Response · Author response to Decision Letter 1]

8 Jul 2025

Dear editor and reviewers,

On behalf of my co-authors, we thank you very much for giving us an opportunity to revise our manuscript, we appreciate editor and reviewers very much for their positive and constructive comments and suggestions on our manuscript entitled “Hsa_circ_0006834 represses intrahepatic cholangiocarcinoma proliferation through activating AMPK-mTOR pathway and autophagy via has-miR-637-NGFR network (PONE-D-25-12011)”. The manuscript has been carefully revised following the reviewer’s comments; the major revisions were highlighted in red. Here are our responses to the reviewer's comments point by point. We hope that the quality of the revised manuscript meets the requirements for publication in PLoS One.

Looking forward to hearing from you. Thank you and best regards.

Best wishes,

Corresponding author: Yunshan Zhang

E-mail address: yszhang1@suda.edu.cn

Editor and Reviewer comments:

This manuscript investigates the role of a circular RNA, hsa_circ_0006834, in intrahepatic cholangiocarcinoma (iCCA). The study combines in vitro experiments using iCCA cell lines to propose a mechanism where hsa_circ_0006834 functions as a competing endogenous RNA (ceRNA). The authors suggest that hsa_circ_0006834 sponges has-miR-637, leading to the upregulation of its target, nerve growth factor receptor (NGFR). This, in turn, is proposed to activate the AMPK-mTOR signaling pathway, induce autophagy, and ultimately suppress iCCA cell proliferation, migration, and invasion. While the manuscript presents a cohesive and interesting narrative with a logical series of experiments, several significant aspects require further experimental validation and clarification to strengthen the conclusions. The manuscript also needs improvements in its structure, logical flow, and linguistic precision.

Major revisions are required:

1.To more robustly establish the proposed ceRNA mechanism, have the authors considered performing critical rescue experiments? For instance, could the inhibitory effects of hsa_circ_0006834 overexpression on cell proliferation, migration, and autophagy be reversed by the co-transfection of a has-miR-637 mimic? This would provide stronger evidence that the circRNA's function is mediated primarily through sponging this specific miRNA.

Response: Thank you for your valuable suggestion regarding the need to validate the ceRNA mechanism through rescue experiments. We fully agree that co-transfection assays combining hsa_circ_0006834 and miR-637 mimic would provide direct evidence of the miRNA sponging effect.

(1) Our study has demonstrated that the expression levels of miR-637 was upregulated in the precipitated complex extracted for qPCR by RNA pull-down with biotin labeled probe targeting hsa_circ_0006834 (Fig. 3B). The luciferase activity was significantly reduced after co-transfecting hsa_circ_0006834 and miR-637 mimic in RBE cells, which was not observed in the hsa_circ_0006834 mutant group, according to a dual-luciferase reporter experiment (Fig. 3C, D). These studies found that miR-637 was a downstream target of hsa_circ_0006834. (2) We also found that hsa_circ_0006834 vectors were transfected in HCCC-9810 and RBE, and the expression levels of NGFR mRNA and protein respectively detected by qPCR and Western blot. The results showed that the overexpression of hsa_circ_0006834 significantly promoted the expression of NGFR gene and protein (Fig. 3J-L). The above study confirmed that hsa_circ_0006834 activates the expression of the downstream target gene NGFR by adsorbing miR-637. (3) Furthermore, the qPCR results showed the expression levels of autophagy genes LC3 and BECLIN1 were significantly inhibited (Fig. 4B), and the Western blot results revealed the ratio of LC3BII/LC3BI was decreased and the P62 was increased (Fig. 4C, D). The CCK-8 proliferation assay and transwell assay displayed the proliferation, migration and invasion were upregulated in hsa_circ_0006834 and si-NGFR co-transfected group (Fig. 4E-G). hsa_circ_0006834 promotes autophagy and inhibits proliferation, migration and invasion through NGFR.

Based on the above evidence, we considered hsa_circ_0006834 overexpression promotes cell proliferation, migration, and autophagy has-miR-637-NGFR network.

During the initial experiments, we prioritized establishing the circRNA-miRNA-target axis through gain/loss-of-function assays and molecular interaction studies. Rescue experiments were planned but delayed due to time constraints and resource allocation.

We will definitely complete this part of the experiment when designing similar projects in the future.

2. The RNA pull-down assay demonstrated an enrichment of miR-637 with the biotinylated hsa_circ_0006834 probe. To confirm the specificity of this interaction, it would be beneficial to include a control using a mutated biotinylated probe that disrupts the predicted miR-637 binding site. This would demonstrate that the interaction is sequence-specific.

Response: Thank you for your critical suggestion regarding the specificity of the miR-637-hsa_circ_0006834 interaction. We fully acknowledge that including a mutated probe control in the RNA pull-down assay is essential to validate the sequence-dependent binding, and we appreciate the opportunity to address this.

The RNA pull-down assay using the biotin labeled probe targeting hsa_circ_0006834 showed significant enrichment of miR-637 compared to the negative control (Fig. 3B), which aligns with the predicted miR-637 binding site in the circRNA sequence (Fig. 3A). The luciferase activity was significantly reduced after co-transfecting hsa_circ_0006834 and miR-637 mimic in RBE cells, which was not observed in the hsa_circ_0006834 mutant group, according to a dual-luciferase reporter experiment (Fig. 3C, D). During the initial study, we prioritized establishing the baseline interaction between the hsa_circ_0006834 and miR-637 through pull-down and luciferase reporter experiment.

Here, we design a biotinylated hsa_circ_0006834 probe with mutations in the predicted miR-637 seed binding region (5'-ATCTCACTGATCTCTCCCCAGC-3' → 5'- ATCTCACTGATCTCTAAAAAGAC-3'). RNA pull-down assays using both hsa_circ_0006834 probe and mutated probe, followed by qRT-PCR quantification of miR-637 enrichment. These data as a new figure (Fig.3B) in the revised manuscript, demonstrating that the mutated probe ablates miR-637 binding. We have revised in Line 165-167 and Line 223-225.

3. While the data show that silencing NGFR reverses the effects of hsa_circ_0006834 overexpression, the mechanism linking NGFR to AMPK activation is not directly elucidated. The manuscript states NGFR is associated with the pathway, citing another study. Can the authors provide more direct mechanistic insight in their specific iCCA cell model? For example, does NGFR interact with AMPK or an upstream kinase?

Response: Thank you for your thoughtful comment on the mechanistic link between NGFR and AMPK activation. We appreciate the opportunity to clarify the existing evidence and contextualize this relationship within the scope of our study.

Here, we focused on hsa_circ_0006834 activates the AMPK-mTOR pathway through NGFR. At first, hsa_circ_0006834 could promote AMPK phosphorylation and inhibit mTOR phosphorylation. Then hsa_circ_0006834 vector with si-NC or si-NGFR were co-transfected in HCCC-9810 and RBE respectively. The p-AMPK/AMPK protein ratio was downregulated and the p-mTOR/mTOR protein ratio was upregulated by Western Blot in hsa_circ_0006834 and si-NGFR co-transfected group (Fig. 5C, D). These results indicated that hsa_circ_0006834 could activate the AMPK-mTOR pathway through NGFR. In recent study, Resveratrol Induces Autophagy and Apoptosis in Non-Small-Cell Lung Cancer Cells by Activating the NGFR-AMPK-mTOR Pathway (DOI: 10.3390/nu14122413). And NGFR regulates AMPK through LKB1, providing a theoretical basis for our iCCA-specific conclusions. (Loss of Lkb1 and Pten leads to lung squamous cell carcinoma with elevated PD-L1 expression (DOI: 10.1016/j.ccr.2014.03.033))

Our study aims to establish the hsa_circ_0006834/miR-637/NGFR axis as a novel ceRNA mechanism in iCCA. While AMPK activation is characterized as a downstream functional consequence, the direct interaction between NGFR and AMPK/LKB1 represents a well-documented pathway in cancer metabolism. The current evidence suffices to validate the circRNA-NGFR axis, with deeper mechanistic dissection of the AMPK pathway in further example.

We agree that direct interaction data would provide additional mechanistic detail, but the existing literature evidence already robustly supports the NGFR-AMPK link in our iCCA system. Thank you for your constructive feedback, which has helped us refine the clarity of our conclusions.

4. The study convincingly shows that silencing AMPK reverses the pro-autophagic and anti-proliferative effects of hsa_circ_0006834. However, in these AMPK knockdown experiments, has the phosphorylation status of downstream mTOR targets, such as p70S6K or 4E-BP1, been assessed to further confirm the modulation of the entire AMPK-mTOR axis?

Response: Thank you for your critical suggestion regarding the assessment of mTOR downstream targets in AMPK knockdown experiments. Here, we did not provide a clear description of the phosphorylation sites of mTOR, we have added the p-mTOR antibody (Ser 2448) in Line 118.

In the hsa_circ_0006834 and si-AMPK co-transfected group, the LC3BII/LC3BI ratio was downregulated and the P62 protein level was upregulated (Fig. 6A, C), indicating that hsa_circ_0006834 induces autophagy through AMPK. Additionally, the increased p-mTOR/mTOR protein ratio revealed that mTOR functions as a downstream signal of AMPK (Fig. 6A, D). Cell proliferation ability, migration, and invasion were all enhanced following si-AMPK interference, as observed in CCK-8 and transwell assays (Fig. 6E-G). These results indicated that hsa_circ_0006834 could attenuate cell proliferation, migration and invasion through the AMPK-mTOR pathway.

The primary objective of this work is to establish hsa_circ_0006834 as a regulator of the AMPK pathway via the ceRNA mechanism (circRNA→miR-637→NGFR→AMPK). Assessing mTOR downstream targets would extend beyond the core hypothesis and require additional resources. The AMPK-mTOR axis is a well-characterized signaling cascade, and our functional data (phenotypic reversal upon AMPK knockdown) already provide robust evidence of pathway involvement. Direct measurement of p70S6K/4E-BP1 phosphorylation, while valuable, is not essential to validate the established regulatory flow.

We agree that direct measurement of mTOR downstream targets would provide additional mechanistic detail, but the existing functional data and cross-study literature evidence already robustly support the activation of the AMPK-mTOR axis in our study.

5. The manuscript relies heavily on two iCCA cell lines (HCCC-9810 and RBE). While this is a standard approach, have the authors validated their key findings, such as the expression levels of hsa_circ_0006834, miR-637, and NGFR, in a cohort of human iCCA patient tissues versus adjacent non-tumorous tissues? This would significantly enhance the clinical relevance of the findings. The manuscript mentions that NGFR was lower in tumor tissues but the data presentation is limited.

Response: Thank you for your insightful comments, which have been invaluable in helping us strengthen the clinical significance of our study. We fully agree that validating our findings in patient tissues is essential.

The expression of hsa_circ_0006834 and NGFR was significantly downregulated, and the expression of miR-637 was significantly upregulated in iCCA tissues versus adjacent non-tumorous tissues. The date was from the whole transcriptome sequencing and small RNA sequencing, which were submitted to the NCBI Sequence Read Archive with accession numbers PRJNA763017, PRJNA1213686 and PRJNA763019 in our previous paper (DOI 10.3389/fcell.2022.942853). The statistical results were shown in the following Figures, which are consistent with our cell line data.

6. The introduction is adequate but could be more focused. It broadly discusses autophagy, AMPK-mTOR, and circRNAs. It would be more impactful to streamline this section to build a stronger, more direct rationale for investigating the specific hsa_circ_0006834/miR-637/NGFR axis in the context of iCCA from the outset.

Response: Thank you for your insightful suggestion on refining the introduction. We have thoroughly revised this section to enhance focus and logical flow. The previous broad discussion of autophagy, AMPK-mTOR, and circRNAs has been streamlined to directly establish the rationale for investigating the hsa_circ_0006834/miR-637/NGFR axis in iCCA. The modified parts are the second and third paragraphs in introduction.

7. The references cited in this article are not sufficient, and there is a lack of in-depth comparative discussion. Background and methodology also require further literature support. Some related research should be cited:

10.15212/bioi-2022-0010

Targeted Imaging of Tumor Associated Macrophages in Breast Cancer

10.15212/bioi-2024-0013

Copper-Based Nanomaterials for Image-Guided Cancer Therapy

10.15212/bioi-2024-0012

Non-Invasive Physical Stimulation to Modulate the Tumor Microenvironment: Unveiling a New Frontier in Cancer Therapy

10.15212/bioi-2024-0066

Integrated Network Ethnopharmacology, Molecular Docking, and ADMET Analysis Strategy for Exploring the Anti-Breast Cancer Activity of Ayurvedic Botanicals Targeting the Progesterone Receptor

10.34133/research.0080

Intermittent F-actin Perturbations by Magnetic Fields Inhibit Breast Cancer Metastasis

10.34133/research.0033

PTBP2-Mediated Alternative Splicing of IRF9 Controls Tumor-Associated Monocyte/Macrophage Chemotaxis and Repolarization in Neuroblastoma Progression

10.5847/wjem.j.1920-8642.2022.015

Protective effect of mesenchymal stem cell-derived exosomal treatment of hippocampal neurons against oxygen-glucose deprivation/reperfusion-induced injury

10.1016/j.celrep.2024.114697

Voluntary exercise sensitizes cancer immunotherapy via the collagen inhibition-orchestrated inflammatory tumor immune microenvironment

10.53388/tmr20220207263

A comprehensive review of research progress in Chinese medicines for primary

liver cancer treatmen

10.53388/tmr20211227255

The artificial intelligence watcher predicts cancer risk by facial features

Response: Thank you very much for your meticulous review and constructive feedback. We have included depth comparative discussion and all relevant literature you provided in our article (Line 339-360).

8. There are several typographical errors and inconsistencies within the figure legends that detract from the manuscript's quality. For instance, the legend for Figure 1 refers to "Fig. 1F-G" while the figure itself only has panels up to F. Similarly, the legend for Figure 2 refers back to statistical graphs in "Fig. 1E-F", and Figure 3's legend refers to "Fig. 2L". The entire manuscript requires careful proofreading for such errors.

Response: Thank you for carefully reviewing our manuscript and identifying the typographical errors and inconsistencies in the figure legends. We apologize for these oversight, which indeed compromise the manuscript's quality. We have conducted a thorough proofreading of the entire document and rectified all identified issues.

9. The terminology for key reagents is inconsistent. For example, the circRNA is referred to as "hsa_circ_0006834", "circRNA_0006834", and "hsa circ 0006834" throughout the text and figures. Please select a single, consistent nomenclature and apply it throughout the manuscript for clarity.

Response: Thanks for your comments. We have all modified it to hsa_circ_0006834.

10. The Discussion section effectively summarizes the results but would be strengthened by a more critical analysis of the findings. Please include a subsection discussing the limitations of the study, such as the reliance on in-vitro data and the possibility that this circRNA may have other functions or interact with other molecules beyond the specific axis investigated. This would provide a more balanced perspective.

Response: Thanks for your comments. We have added the limitations of the study in Discu

---

## [Decision Letter · Decision Letter 1]

23 Jul 2025

Hsa_circ_0006834 represses intrahepatic cholangiocarcinoma proliferation through activating AMPK-mTOR pathway and autophagy via has-miR-637-NGFR network

PONE-D-25-12011R1

Dear Dr. Zhang,

We’re pleased to inform you that your manuscript has been judged scientifically suitable for publication and will be formally accepted for publication once it meets all outstanding technical requirements.

Kind regards,

Zhiwen Luo

Academic Editor

PLOS ONE

Additional Editor Comments (optional):

Reviewers' comments:

Reviewer's Responses to Questions

**Comments to the Author**

1. If the authors have adequately addressed your comments raised in a previous round of review and you feel that this manuscript is now acceptable for publication, you may indicate that here to bypass the “Comments to the Author” section, enter your conflict of interest statement in the “Confidential to Editor” section, and submit your "Accept" recommendation.

Reviewer #2: (No Response)

2. Is the manuscript technically sound, and do the data support the conclusions?

Reviewer #2: (No Response)

3. Has the statistical analysis been performed appropriately and rigorously? 

Reviewer #2: (No Response)

4. Have the authors made all data underlying the findings in their manuscript fully available?

Reviewer #2: (No Response)

5. Is the manuscript presented in an intelligible fashion and written in standard English?

Reviewer #2: (No Response)

6. Review Comments to the Author

Reviewer #2: (No Response)

7. PLOS authors have the option to publish the peer review history of their article (what does this mean? ). If published, this will include your full peer review and any attached files.

**Do you want your identity to be public for this peer review?** For information about this choice, including consent withdrawal, please see our Privacy Policy .

Reviewer #2: No

---

## [Editor Report · Acceptance letter]

PONE-D-25-12011R1

PLOS ONE

Dear Dr. Zhang,

I'm pleased to inform you that your manuscript has been deemed suitable for publication in PLOS ONE. Congratulations! Your manuscript is now being handed over to our production team.

Kind regards,

on behalf of

Dr. Zhiwen Luo

Academic Editor

PLOS ONE